Bioinformatics analysis of the structural and evolutionary characteristics for toll-like receptor 15

Wang Jinlan 1
Zhang Zheng 2 zhzhang.sdu@gmail.com
Chang Fen 1
Yin Deling 3 4 delingyin@yahoo.com
1 Institute of Developmental Biology, School of Life Science, Shandong University , Jinan , China
2 State Key Laboratory of Microbial Technology, School of Life Science, Shandong University , Jinan , China
3 School of Pharmacy, Central South University , Changsha , China
4 Department of Internal Medicine, College of Medicine, East Tennessee State University , Johnson, TN , USA
D’Acquisto Fulvio
Electronic publication date: 2016 May 25
Publication date: 2016
Volume: 4
Electronic Location ID: e2079
Received 2016 Feb 26; Accepted 2016 May 3
Copyright: ©2016 Wang et al.
Copyright year: 2016
Copyright holder: Wang et al.
License: This is an open access article distributed under the terms of the Creative Commons Attribution License, which permits unrestricted use, distribution, reproduction and adaptation in any medium and for any purpose provided that it is properly attributed. For attribution, the original author(s), title, publication source (PeerJ) and either DOI or URL of the article must be cited.
License URL: https://creativecommons.org/licenses/by/4.0/

Keywords: Toll-like receptor 15, Innate immunity, Structural characteristics, Protein–protein interactions, Molecular evolution

Funding: National 973 Research Project 2014CB542400 National Natural Science Foundation of China 81570454 This work was supported in part by the National 973 Research Project (No. 2014CB542400) and National Natural Science Foundation of China (No. 81570454). The funders had no role in study design, data collection and analysis, decision to publish, or preparation of the manuscript.

==============================
Toll-like receptors (TLRs) play important role in the innate immune system. TLR15 is reported to have a unique role in defense against pathogens, but its structural and evolution characterizations are still poorly understood. In this study, we identified 57 completed TLR15 genes from avian and reptilian genomes. TLR15 clustered into an individual clade and was closely related to family 1 on the phylogenetic tree. Unlike the TLRs in family 1 with the broken asparagine ladders in the middle, TLR15 ectodomain had an intact asparagine ladder that is critical to maintain the overall shape of ectodomain. The conservation analysis found that TLR15 ectodomain had a highly evolutionarily conserved region on the convex surface of LRR11 module, which is probably involved in TLR15 activation process. Furthermore, the protein–protein docking analysis indicated that TLR15 TIR domains have the potential to form homodimers, the predicted interaction interface of TIR dimer was formed mainly by residues from the BB-loops and αC-helixes. Although TLR15 mainly underwent purifying selection, we detected 27 sites under positive selection for TLR15, 24 of which are located on its ectodomain. Our observations suggest the structural features of TLR15 which may be relevant to its function, but which requires further experimental validation.

Introduction

Innate immunity stands on the first line of immune defense against pathogens. Toll-like receptors (TLRs) are a major class of pattern recognition receptors in innate immunity, they recognize a variety of highly conserved pathogen-associated molecular patterns (PAMPs) in pathogens to initiate an innate immune response and prime the adaptive immune system (Akira & Takeda, 2004). TLR is characterized by the presence of an ectodomain that is involved in recognizing ligands and an intracellular Toll/IL-1 receptor-like (TIR) domain that mediates signaling (Kang & Lee, 2011). The ectodomain of TLR contains a large number of leucine-rich repeat (LRR) modules and is generally bent into a characteristic horseshoe-shaped structure (Botos, Segal & Davies, 2011).

So far, 27 types of TLRs have been identified in vertebrates (Wang et al., 2015a; Wang et al., 2015b), the ligand specificities of some TLRs have been clarified. TLR2 is able to form a heterodimer with TLR1, TLR6 or TLR10 to detect microbial lipopeptides (Guan et al., 2010; Jin et al., 2007; Kang et al., 2009); TLR3 and TLR22 detect double-stranded RNA (Liu et al., 2008; Matsuo et al., 2008); TLR4 recognizes to lipopolysaccharides from Gram-negative bacteria (Park et al., 2009); TLR5 binds to bacterial flagellin (Yoon et al., 2012); TLR7, TLR8, and TLR13 bind to single-stranded RNA (Heil et al., 2004; Shi et al., 2011; Tanji et al., 2015), TLR9 and TLR21 recognize unmethylated CpG-containing DNA (Brownlie et al., 2009; Keestra et al., 2010; Ohto et al., 2015; Yeh et al., 2013), and TLR11 and TLR12 respond to profilin (Koblansky et al., 2013; Yarovinsky et al., 2005).

TLR15 can be identified in avian and some reptilian species (Alcaide & Edwards, 2011; Boyd et al., 2012). TLR15 was considered to play a constitutive role in the immune defense of chickens (Higgs et al., 2006). The expression of TLR15 was strongly up-regulated after Salmonella enterica infection (Higgs et al., 2006; Hu et al., 2016; Shaughnessy et al., 2009). TLR15 was reported to have a unique activation mechanism, where the cleavage of TLR15 ectodomain by secreted microbial proteases results in its activation (De Zoete et al., 2011). A later study showed that TLR15 recognized a yeast-derived agonist that was heat labile and inhibited by PMSF (Boyd et al., 2012). The diacylated lipopeptide from Mycoplasma synoviae was reported to mediate TLR15-dependent innate immune responses (Oven et al., 2013).

Benefiting from the recently rapid increasing of genome data, a large number of TLR sequences are determined (Zhang et al., 2014). In the current work, we identified 57 completed TLR15 genes from the genomes in vertebrates, investigated the phylogenetic relationships, structural and evolutionary characterizations using these sequences, and predicted the dimeric interaction of TLR15 TIR domains.

Materials and Methods

Phylogenetic analysis

The coding sequences of vertebrate TLR genes were retrieved from NCBI (GenBank) and Ensembl databases (Benson et al., 2013; Cunningham et al., 2015). All the partial sequences (<1,800 bp) and pseudogene sequences were excluded. A multiple sequence alignment of the TLR proteins was constructed with MAFFT (FFT-NS-i, BLOSUM62) (Katoh & Standley, 2013), and a phylogenetic tree was calculated from it with PhyML using the LG substitution model and four substitution rate categories (Guindon et al., 2010; Le & Gascuel, 2008). Branch support was calculated with an approximate likelihood ratio tests (aLRT SH-like) (Anisimova & Gascuel, 2006). The phylogenetic tree was visualized with MEGA 6 (Tamura et al., 2013).

Structural elements analysis

SignalP, SMART, and TMHMM were used to identify the signal peptide, ectodomain, transmembrane region, and intracellular TIR domain of TLR15 (Krogh et al., 2001; Letunic, Doerks & Bork, 2015; Petersen et al., 2011). The delimitation of each LRR module in the TLR15 ectodomain was determined with the LRRfinder software (Offord, Coffey & Werling, 2010). The delimitations of chicken TLR15 were used as the reference for the remaining species.

Structural modeling

I-TASSER was used to model the structure of the ectodomain and TIR domain of TLR15 based on a threading approach (Yang et al., 2015). I-TASSER is a hierarchical method for protein structure prediction. Structural templates were first identified from the PDB by the multiple-threading program LOMETS; then, full-length models were constructed by iterative template fragment assembly simulations. The modeled structure was displayed by PyMol (Schrödinger, LLC).

Protein–protein docking analysis

The two modeled structures of chicken TLR15 TIR domains were submitted as target 1 and target 2, respectively, to PRISM protein–protein docking server (Baspinar et al., 2014; Tuncbag et al., 2011). PRISM predicts possible interactions, and how the interaction partners connect structurally, based on geometrical comparisons of the template structures and the target structures.

Residue conservation analysis

A multiple sequence alignment of the TLR15 proteins was constructed with MAFFT. The evolutionary conservations of amino acid residue positions in the TLR15 sequences was estimated by using ConSurf algorithm (Ashkenazy et al., 2010). The JTT substitution matrix was used and computation was based on the empirical Bayesian paradigm. The conservation scale was defined from the most variable amino acid positions (grade 1) which were considered as rapidly evolving, to the most conservative amino acid positions (grade 9) which were considered as slowly evolving. The sequence and modeling structure of chicken TLR15 were used to show the nine-color conservation grades.

Codon-based analyses of positive selection

A multiple sequence alignment of the nucleic acid sequences of the TLR15 genes, based on their codons, was established with TranslatorX (Abascal, Zardoya & Telford, 2010). The ratio of the number of nonsynonymous substitutions per nonsynonymous site (dN) to the number of synonymous substitutions per synonymous site (dS), dN/dS, an indicator of the selective pressure acting on a protein-coding gene, at the TLR15 locus and the corresponding 95% confidence interval were calculated with the Datamonkey web server (Delport et al., 2010).

The SLAC, FEL, REL, and FUBAR methods were implemented in Datamonkey to explore the evidence of positive selection acting on the individual codon of the TLR15 sequences (Kosakovsky Pond & Frost, 2005; Murrell et al., 2013). To minimize the overestimation of the positively selected codons, codons with p values < 0.1 for SLAC and FEL, with Bayes Factor > 50 for REL, and with posterior probabilities > 0.9 for FUBAR were considered as candidates to be under positive selection.

Results

TLR15 is phylogenetically closely related to family 1

We obtained all the known vertebrate full-length TLR gene sequences and constructed their phylogenetic relationships based on maximum likelihood method (Fig. 1A). TLR15 clusters into an individual clade and is closely related to the TLRs in family 1 (TLR1, TLR2, TLR6, TLR10, TLR14, TLR18, TLR25 and TLR27) on the phylogenetic tree. This is consistent with the early finding (Roach et al., 2005).

Figure 1 Phylogenetic analysis of TLR15 and the other vertebrate TLRs.

(A) A large unrooted tree of all the known vertebrate TLRs. Maximum likelihood tree was constructed based on the full-length sequences of TLRs. Six TLR families are labeled in the tree. The clades of family 1 are shown in red and the clades of TLR15s are shown in bold. (B) Amplified TLR15 clades of the large unrooted tree. The support value at each branching point is shown. Its robustness was estimated with an aLRT SH-like method.

We identified 57 completed TLR15 genes through the phylogenetic analysis (Fig. 1B, Table S1). These TLR15 sequences are derived from 54 avian and 3 reptilian species (Burmese python, Chinese alligator, and American alligator). TLR15 exists in these species as a single gene copy in each species’ genome. Also, the phylogenetic tree showed that TLR15 has evolved following the phylogeny of species. These results indicate that TLR15 is possibly subjected to strong selection constraints.

TLR15 ectodomain possesses an intact asparagine ladder

TLR15 generally include 860∼880 amino acids. Motif prediction showed that TLR15 included the signal peptide, ectodomain, transmembrane region, and intracellular TIR domain. The structure of chicken TLR15 ectodomain was modeled with threading approach (Fig. 2A). The estimated TM-score (0.72 ± 0.11) for this modeled structure showed that it was acceptable. TLR15 ectodomain contains a LRRNT, a LRRCT, and 19 LRR modules, which is identical to the number of LRR modules of the known TLRs in family 1. The LRR3 module of TLR15 is quite long, for example, the LRR3 of chicken TLR15 has 99 amino acids that far exceed the lengths of common LRR modules (∼24 amino acids).

Figure 2 Comparison between the asparagine ladders in the ectodomains of TLR15 and the other TLRs in family 1.

(A) TL15 has an intact asparagine ladder in the ectodomain. The model is chicken TLR15 ectodomain. There was a predicted long loop between LRR3 and LRR4 modules, in which the conserved “LxxLxLxxNxL” motif was not found. (B) The asparagine ladders of the other TLRs in family 1 are broken in the middle. The crystal structure of human TLR2 ectodomain (PDB code: 2Z7X) is displayed as an example. The ectodomain structures are shown in cartoon mode. The residues in the asparagine ladder position (cyan) are shown by sphere mode. The identifying numbers of the 19 canonical LRR, LRRNT and LRRCT modules are labeled.

The LRR module is characterized by a conserved “LxxLxLxxNxL” motif on the concave surface. Among them, the conserved asparagine plays important role in maintaining the overall shape of TLR ectodomain by forming a continuous hydrogen-bond network (called as asparagine ladder) with backbone carbonyl oxygens of neighboring LRR modules (Kang & Lee, 2011). The asparagines can be substituted by other amino acids that are able to donating hydrogens, such as threonine, serine, and cysteine. We analyzed the asparagine ladder in the ectodomain of TLR15. The results showed that the asparagine ladder was intact throughout the TLR15 ectodomain, suggesting that TLR15 possesses an intact and continuous hydrogen-bond network (Fig. 2A).

Although both TLR15 and the TLRs in family 1 contain 19 LRR modules, the known crystal structures indicate that the asparagine ladders of the ectodomains of TLRs in family 1 are broken in the middle (Fig. 2B). TLR15 ectodomain more approximates to those TLRs with intact asparagine ladders in the ectodomains, for example, TLR3, TLR5, TLR7, and TLR21 (Table S2). Therefore, the ectodomain of TLR15 is obviously structurally different from the TLRs in family 1.

The convex surface of TLR15 ectodomain has a highly evolutionarily conserved region

We calculated the evolutionary conservation scores of all residue positions of TLR15 based on the phylogenetic relationships between homologous sequences (Fig. 3). The conservation scale was defined from the most variable amino acid positions, which were considered as rapidly evolving, to the most conservative amino acid positions, which were considered as slowly evolving. Further, the mean evolutionary conservation score of each module in TLR15 was also calculated (Fig. 4).

Figure 3 Evolutionary conservation s of amino acid positions displayed in the sequence of TLR15.

The conservation scale was defined from the most variable amino acid positions (grade 1, colored turquoise) to the most conservative amino acid positions (grade 9, colored maroon). Positions, for which the inferred conservation level was assigned with low confidence, are marked with light yellow. The sequence of chicken TLR15 was used to show the nine-color conservation grades. The signal peptide, the predicted LRR modules of the ectodomain, transmembrane region (TM), and intracellular domain (TIR) for TLR15 are labeled. The residues in asparagine ladder positions in the concave surface of each LRR module, the sites under positive selection, and the residues involved in the homodimeric interaction of TLR15 TIR domains are marked with solid gray circles, solid red circles, and solid green circles under the sequence, respectively.

Figure 4 Mean evolutionary conservation of each module of TLR15.

SP, Signal peptide; NT, LRRNT module; CT, LRRCT module; TM, transmembrane region; TIR, intracellular domain. The different LRR modules of the ectodomain are represented by their identifying numbers. The lowest score represents the most conserved position in a protein. The error bars represent the standard error of the mean (SEM).

The results showed that the majority of amino acid positions of intracellular TIR domain were highly evolutionarily conserved. However, the average evolutionary conservation of different LRR module in the ectodomain had large differences. Most of LRR3 module was composed of highly variable positions. The loop of convex surface in LRR9 module also includes a number of highly variable positions. The average evolutionary conservations of LRR18, LRR19, and LRRCT modules are very high, this could be adapted to their function participating in the dimerization of C-terminal region of TLR15 ectodomain. The average evolutionary conservations of LRR10-13 modules are also very high, indicating that they are possibly related to the function of the ectodomain sensing pathogens.

We further displayed the evolutionary conservations of amino acid residue positions in the ectodomain of TLR15 using the modeled structure (Fig. 5). The asparagine ladder with high evolutionary conservation is located on the ascending lateral surface, which contains loops connecting the β-strand of the concave surface to the convex surface and participates in the dimerization of TLRs (Botos, Segal & Davies, 2011). Compared to the ascending lateral surface, its opposite side, the descending lateral surface, includes fewer evolutionarily conserved positions. Interestingly, we found a highly conserved region on the convex surface of TLR15 ectodomain, which is exactly located on the LRR11 module. For chicken TLR15, the highly conserved region on the convex surface of its LRR11 module ranges from 397th to 406th amino acid residues (SIVELPEWFA). The high conservation of this region across species might suggest that it is involved in ligand recognition.

Figure 5 Surface evolutionary conservation of TLR15 ectodomain.

A highly evolutionarily conserved region on the convex surface of TLR15 ectodomain is labeled with dashed yellow circle. The modeling structure of chicken TLR15 was used to show the conservation. The surfaces are colored according to ConSurf results: the most variable (turquoise) to the most conserved (maroon).

TLR15 TIR domains are able to form homodimers

We further studied TLR15 intracellular TIR domain that is responsible for the signal transduction. The structure of TIR domain (from Pro706 to Thr848) of chicken TLR15 was modeled with threading approach. This modeled structure mainly contains a central three-stranded parallel β-sheet surrounded by five α-helices (Fig. 6A).

Figure 6 Prediction of TLR15 TIR domain homodimeric interaction.

(A) Cartoon figure of TLR15 TIR homodimeric interaction predicted through docking calculations. The left monomeric structure is colored according to the conservation score of each residue position, while the N-terminus to the C-terminus of the right one is colored from blue to red. The homodimerization interface has been split and rotated to show the involved residues. (B) Inter-residues interaction in potential dimerization interfaces. The interaction partners are connected by broken lines. The modeled structure of chicken TLR15 TIR domain was used for docking analysis. The residues are numbered according to the chicken TLR15 sequence.

Protein–protein docking analysis showed that TLR15 TIR domain was able to form the homodimer with itself (Fig. 6A). The dimeric interface is mainly formed by the residues from the BB-loops and αC-helixes. The fifteen residues of TLR15 TIR domain were predicted to participate in its homodimerization, the five of them are located on the BB-loops and six of them are located on the αC-helixes (Fig. 6B). Cys777 in αC-helix at one monomer (chTLR15) probably formed disulfide bond with the same residue from the second molecule (chTLR15’). The amino acid residue positions involved in the interaction of TIR homodimer are highly evolutionarily conserved among 57 TLR15 sequences (Fig. 6B). The fourteen of fifteen residue positions involved in the homodimeric interaction have an evolutionary conservation score greater than or equal to the grade 8.

We compared the predicted dimerization of TLR15 with those of the TLRs in family 1. The multiple sequence alignment showed that there were 30% identical amino acid residues among the TIR domains of chicken TLR15, human TLR1, human TLR2, human TLR6, and human TLR10 (Fig. 7). The known five functionally critical residues in TLR2 TIR domain are conserved among different TLRs (Tao et al., 2002), which correspond to Phe743, Pro745, Cys777, Leu781, and Lys812 in chicken TLR15. The homodimer interface of TLR6 TIR domains is considered to be formed mainly by the residues from αC-helixes (Jang & Park, 2014), the six of its eight interaction residues was also found in the interaction of TLR15 TIR domain homodimer. The homodimer interface of TLR10 TIR domains was reported to mainly contain residues from the BB-loops and αC-helixes (Nyman et al., 2008). The nine of its eleven interaction residues was also detected in the interaction of TLR15 TIR domain homodimer. Comparatively speaking, the interaction mode of TLR15 TIR homodimer is more similar to that of TLR10.

Figure 7 Alignment of representative TIR domain sequences from different TLRs.

The surface residues involved in the homodimeric interaction detected by protein–protein docking analysis for TLR15 are shaded in green. The surface residues on TLR6 and TLR10 involved in TIR–TIR interaction are shaded in pink and yellow, respectively. The surface residues on TLR2 that have been known to be critical for the TLR signaling are shaded in light blue. The elements of secondary structures are labeled above the sequence. Consistent with the previous work of TIR domains, the loops are named by the strands and helices that they connect.

Twenty-seven sites under positive selection were detected for TLR15

The estimated dN/dS value for the TLR15 locus is 0.318 (95% CI [0.308–0.328]), suggesting a clear excess of synonymous over nonsynonymous substitutions at TLR15 locus. However, numerous sites under positive selection were found in TLR15 with the codon-based maximum likelihood methods (Table 1). The SLAC, FEL, REL, and FUBAR methods identified statistically significant positive selection in 24, 29, 22, 14 amino acid sites across TLR15s, respectively. To identify robust candidate sites under positive selection, only 27 codons with evidence of positive selection identified by at least two methods were considered, which account for 3.1% of the total sites in TLR15.

Table 1 Tests for positive selection of TLR15s.

Methods	Sites under positive selectiond	
SLACa	19, 26, 33, 89, 102, 120, 136, 169, 188, 197, 253, 262, 283, 326, 333, 376, 407, 430, 517, 615, 617, 621, 716, 862	
FELa	19, 26, 89, 102, 114, 120, 136, 169, 185, 188, 197, 253, 262, 326, 333, 353, 360, 376, 407, 430, 517, 544, 597, 615, 617, 621, 643, 716, 862	
RELb	89, 114, 120, 132, 169, 197, 253, 262, 333, 337, 376, 407, 410, 430, 452, 517, 544, 615, 617, 650, 655, 862	
FUBARc	26, 102, 136, 185, 197, 253, 326, 333, 353, 360, 430, 615, 617, 621	
Notes.

a Codons with p values < 0.1.

b Codons with Bayes factor > 50.

c Codons with posterior probability > 0.9.

d Those positively selected sites identified by more than one method are underlined. Sites are numbered according to the chicken TLR15 sequence.

These 27 sites under positive selection belong to the most variable positions identified by evolutionary conservation analysis (Fig. 3). The 24 of 27 sites under positive selection are located on TLR15 ectodomain. Among them, the LRR3 module contains seven sites under positive selection. The proline-rich loop on the convex surface of LRR9 module includes two sites under positive selection. There is also a site under positive selection (407th) closely neighboring to the highly evolutionarily conserved region on the convex surface of LRR11 module.

Discussion

Although TLR15 is phylogenetically close to family 1, our modeling structure showed that TLR15 ectodomain was obviously structurally different from the TLRs in family 1. The known crystal structures indicate that the asparagine ladders in the ectodomains of TLRs in family 1 are broken in the middle, thus resulting in the structural transition of their ectodomain in the middle, and further causing the formation of a hydrophobic pocket at the boundary between the LRR11 and LRR12 modules that is responsible for binding to ligands (Jin et al., 2007; Kang et al., 2009). In contrast, the asparagine ladder in the ectodomain of TLR15 is intact, thus indicating that TLR15 possibly does not have the structural transition and also does not form a hydrophobic pocket in the middle of its ectodomain. Therefore, considering the large sequence and structural differences between TLR15 and the TLRs in family 1, TLR15 should be regarded as an individual family.

TLR15 is able to be activated through the cleavage of its ectodomain by secreted virulence-associated fungal and bacterial proteases, and also can be activated by the diacylated lipopeptide from Mycoplasma synoviae (De Zoete et al., 2011; Oven et al., 2013). However, the functional sites on the ectodomain of TLR15 are still unclear. We found a highly evolutionarily conserved region in the convex surface of TLR15 ectodomain by using the information of a large number of the known sequences. We infer that this highly evolutionarily conserved region is probably closely related to the function of TLR15. Interestingly, this highly evolutionarily conserved region in TLR15 ectodomain is located on the convex surface of LRR11 module, whereas the known ligand-binding regions of TLRs in family 1 are also located on the border between the LRR11 and LRR12 modules.

The TIR domains of TLRs are responsible for signal transduction in response to stimulation from pathogens. The formation of TIR domain complex is critical for receptor signaling. The TIR domain interactions mediate the oligomerization of receptor TIR domains, and also mediate the association between the receptor and adapter TIR domains (Xu et al., 2000). We predicted that there was quite strong interaction between TLR15 TIR domain and itself, suggesting that TLR15 may recruit downstream adaptor in the form of a homodimer. The formation of TLR15 TIR domain homodimers can be bolstered by the known ability to signal when only TLR15 is transfected into HEK293 cells (Boyd et al., 2012; De Zoete et al., 2011). In previous structural works, several TIR-dimer interaction modes have been proposed (Jang & Park, 2014; Nyman et al., 2008; Xu et al., 2000). The five residues in TLR2 TIR domain were verified to be functionally important for signaling (Tao et al., 2002). We found that their equivalent residues in TLR15 were involved in the predicted homodimerization of TIR domains. Also, these five residues are highly evolutionarily conserved among TLR15s. In particular, the previous study demonstrated that Pro681 in the BB-loop of TLR2 TIR domain (corresponding to Pro745 in chicken TLR15) mediated the interaction with adaptor MyD88 and further BB-loop was suggested to be the site of adaptor protein recruitment (Xu et al., 2000). The mutation of the equivalent residue in the BB-loop of human TLR10 TIR domain also affects the interaction with adaptor MyD88 (Hasan et al., 2005). Therefore, some residues on the interface of TLR15 TIR domain homodimer also probably participate in the interaction with downstream adapter.

Toll-like receptors are located directly on the host-pathogen interface and might undergo coevolutionary dynamics with their pathogenic counterparts. The dN/dS value showed that TLR15 was evolutionarily conserved, but many positively selected sites were still identified. The previous study identified 8 positively selected sites in the ectodomain among six TLR15 sequences through REL method (Alcaide & Edwards, 2011). We identified 27 robust positively selected sites among 57 full-length TLR15 sequences using four methods. These positively selected sites are mainly located on the ectodomain that is responsible for recognizing the ligands, and are possibly related to the function of TLR15. Therefore, the fact that so many positively selected sites were identified shows that there are coevolutionary dynamics between TLR15 and their pathogenic counterparts.

Conclusions

In this study, we identified 57 completed TLR15 genes from a large number of avian and reptilian genomes. TLR15 is phylogenetically closely related to family 1. Unlike the TLRs in family 1 with the broken asparagine ladders in the middle, TLR15 ectodomain possesses an intact asparagine ladder. The convex surface of TLR15 ectodomain has a highly evolutionarily conserved region, which is probably related to the function of TLR15. We found that TLR15 TIR domains were able to form homodimers in silico. The major contributions to the homodimer interface of TLR15 TIR domains are made by residues from the BB-loops and αC-helixes. Twenty-seven sites under positive selection that are probably associated with function were detected for TLR15. Overall, these findings provide novel insights into the structural and evolutionary characterizations of TLR15.

Supplemental Information

Table S1 Information of the completed TLR15 gene sequences from avian and reptilian species

Click here for additional data file.

Table S2 Amino acid compositions in the asparagine ladder positions of avian TLR ectodomains

Click here for additional data file.

Data S1 57 completed TLR15 gene sequences

Raw data.

Click here for additional data file.

We would like to thank Hui Fu, Jing Liu, and Jing Zhao for providing the useful suggestions on the manuscript.

Additional Information and Declarations

Competing Interests

Author Contributions

Data Availability

The authors declare there are no competing interests.

Jinlan Wang and Zheng Zhang conceived and designed the experiments, performed the experiments, analyzed the data, wrote the paper, prepared figures and/or tables, reviewed drafts of the paper.

Fen Chang performed the experiments, analyzed the data.

Deling Yin conceived and designed the experiments, reviewed drafts of the paper.

The following information was supplied regarding data availability:

The raw data has been supplied as Data S1.

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
