# Peer review of "Bioinformatics analysis of the structural and evolutionary characteristics for toll-like receptor 15"

_PeerJ, doi:10.7717/peerj.2079_

## Round 0.1 · original submission · Major Revisions

It seems to me that both reviewers have done a great job and provided very constructive comments. I am confident that addressing as many points that has been raised as possible will greatly improve the impact of the study.

Reviewer 1 ·

Basic reporting

The manuscript by Wang et al., provides an analysis of TLR15 from a wide range of species and including phylogenetic, interspecies polymorphism analyses and some predicted structural analyses. Although the title suggests that the manuscript deals with the evolution of this TLR the data does not cover this topic effectively. The title also indicates that this TLR has a unique activation mechanism, a statement based entirely on work presented in a single paper (deZoete et al., 2011 PNAS) which reported activation based upon enzymatic cleavage, primarily using Proteinase K to cleave the TLR. Other manuscripts report activation of TLR15 but do not report cleavage, in particular the paper Oven et al., Vet Res 2013 appears to have been completely overlooked, this paper reports TLR15 activation by diacyelated lipopeptide (and not an enzyme). This area is not adequately introduced or discussed.
The abstract mentions self-activation of TLR15 (line 2), which has never been proposed and appears to be a mistake. Similarly it is unclear what massive whole genomes means (line 4). A similar error is made in the discussion.
The abstract also indicates that TLR15 is a member of family 1 within the TLR and this is not strongly supported in any of the papers that offer phylogenetic analyses (e.g. Roach et al. 2005, Boyd et al., 2012, Alcaide and Edwards 2011) or in the work performed within the current manuscript (that does include more examples of TLR15 from different species). TLR15’s lie in a clade that is separate from all other vertebrate TLR. The analyses in the current manuscript uses the whole TLR sequence and this is questionable since the extracellular PRR containing domains are too variable between TLR to effectively align TLRs from distant groups. The work of other groups is also not adequately discussed in the current manuscript.
The abstract also identifies the potential for TLR15 to form TIR domain homodimers this is as expected with most TLR including TLR2 (which forms both homodimers and heterodimers) (note the current manuscript suggests TLR2 only forms heterodimers). (also an issue in the introduction and discussion).
The abstract also indicates sites where positive selection can be identified and does not discuss these in the context of the Alcaide and Edwards 2011 paper.
Introduction last paragraph, what is a partial reptilian species?
Results:
Where the authors discuss intact or broken asparagine ladders the TLR1/2 are used as a comparison but if TLR15 is unrelated (especially in its extracellular domain then why should we expect them to be more like TLR1/2 and not others with intact asparagine ladders, some examples are given in the text (without citation or reference to work done for this manuscript). Indeed this analysis should be done for all avian TLR and then each compared with TLR15.
Similarly to place the work on TLR15 into context then all of the analyses presented in the manuscript should be done on a range of different TLR.
There are comments on identifying a site for proteolytic cleavage without any real justification or discussion in the context of the deZoete et al., 2011 paper. Do the regions identified match each other?
A typo on line 233….pro-life rich rather than proline
Discussion: The opening comment that TLR15 is “traditionally” considered a member of the TLR1 family is simply not true.
The discussion of TLR15 TIR domain homodimers should be bolstered by the known ability to signal when transfected into HEK293 cells (without any other TLR) (deZoete and Boyd papers).
Overall this manuscript contains a series of bioinformatic and protein structure modelling approaches linked to TLR15 and to become a meaningful set of comparisons should include comparisons to all other TLR (and not just TLR1/2 members).

Experimental design

The research question is not entirely clear and it would be much more meaningful if the analyses compared all of the TLR (then it might conclude that TLR15 is unusual or comparable to other TLR).

Validity of the findings

Overall seem fair but based either on bioinformatics or simulation of structures, there was no validation of any of the findings.
With Figure 4 it is unclear what the SEM relate to.
The choices of colors for many of the structural figures (particularly maroon and turquoise) are not the best choices (particularly in the context of people who are red-green colorblind).

·

Basic reporting

No comments. Suggested changes are given below

Experimental design

No comments

Validity of the findings

Because all the findings are based on bioinformatics, conclusions of the function and structure of TLR15 are speculative and the wording should be changed to indicate this.

Additional comments

This article describes an analysis of structural and evolutionary properties of TLR15 based solely on bioinformatics. Thus most of the conclusions are speculative unless verified by experiments and this should be stated in the text. I think it is an overstatement to conclude “these findings help better understand the unique activation mechanism and the molecular
basis for signal transduction of TLR15”. It is more accurate to say that the observations suggest structural features of TLR15 which may be relevant to its function but which require experimental validation. Specific points are:
1. The title could be much better – it is not a comprehensive analysis and is based on bioinformatics and TLR15 should be in the title.
2. Abstract L30 – TIR domains have the potential to form homodimers.
3. Roach 2005 also demonstrated that TLR15 clustered into an individual Clade although most closely related to family 1 – this should be cited.
4. L187. “To chicken TLR15” should be “For chicken TLR15”
5. L248-250. It is not clear what the functional study is. These two sentences need to be reworded.
6.L271-279. TLRs other than those of the family 1 form homodimers – this is not unique to TLR15
7. L274. may recruit downsteam adaptor in the form of a homodimer
8. L276. In a previous study,

---

## Round 0.2 · accepted · Accept

All the points raised by the reviewer have been addressed.